# PromarkerD Predicts Renal Function Decline in Type 2 Diabetes in the Canagliflozin Cardiovascular Assessment Study (CANVAS)

**DOI:** 10.3390/jcm9103212

**Published:** 2020-10-06

**Authors:** Kirsten E. Peters, Jialin Xu, Scott D. Bringans, Wendy A. Davis, Timothy M.E. Davis, Michael K. Hansen, Richard J. Lipscombe

**Affiliations:** 1Proteomics International, Nedlands, WA 6009, Australia; kirsten@proteomics.com.au (K.E.P.); scott@proteomics.com.au (S.D.B.); richard@proteomics.com.au (R.J.L.); 2Medical School, The University of Western Australia, Fremantle Hospital, Fremantle, WA 6959, Australia; wendy.davis@uwa.edu.au; 3Janssen Research and Development, LLC, Spring House, PA 19477, USA; J.X.u34@its.jnj.com (J.X.); M.H.ansen3@its.jnj.com (M.K.H.)

**Keywords:** type 2 diabetes, diabetic nephropathy, renal decline, chronic kidney disease, biomarkers, risk prediction, prognosis

## Abstract

The ability of current tests to predict chronic kidney disease (CKD) complicating diabetes is limited. This study investigated the prognostic utility of a novel blood test, PromarkerD, for predicting future renal function decline in individuals with type 2 diabetes from the CANagliflozin CardioVascular Assessment Study (CANVAS). PromarkerD scores were measured at baseline in 3568 CANVAS participants (*n* = 1195 placebo arm, *n* = 2373 canagliflozin arm) and used to predict incident CKD (estimated glomerular filtration rate (eGFR) <60 mL/min/1.73m^2^ during follow-up in those above this threshold at baseline) and eGFR decline ≥30% during the 4 years from randomization. Biomarker concentrations (apolipoprotein A-IV (apoA4), CD5 antigen-like (CD5L/AIM) and insulin-like growth factor-binding protein 3 (IGFBP3) measured by mass spectrometry were combined with clinical data (age, serum high-density lipoprotein (HDL)-cholesterol, eGFR) using a previously defined algorithm to provide PromarkerD scores categorized as low-, moderate- or high-risk. The participants (mean age 63 years, 33% females) had a median PromarkerD score of 2.9%, with 70.5% categorized as low-risk, 13.6% as moderate-risk and 15.9% as high-risk for developing incident CKD. After adjusting for treatment, baseline PromarkerD moderate-risk and high-risk scores were increasingly prognostic for incident CKD (odds ratio 5.29 and 13.52 versus low-risk, respectively; both *p* < 0.001). Analysis of the PromarkerD test system in CANVAS shows the test can predict clinically significant incident CKD in this multi-center clinical study but had limited utility for predicting eGFR decline ≥30%.

## 1. Introduction

Chronic kidney disease (CKD) is common in people with type 2 diabetes (T2D), with up to one in three people being affected [1]. Among those with diabetes and Stage 3 or 4 CKD (estimated glomerular filtration rate (eGFR) 15–59 mL/min/1.73m^2^) [2], only 25% are aware of the presence of kidney disease [1]. Diabetes is the leading cause of end stage renal disease (ESRD) with 38% of ESRD cases in the United States (US) attributable to diabetes [3]. Diabetes-associated CKD is the 16th leading cause of death in the US, accounting for 40,000 deaths per year [4].

CKD in people with diabetes is commonly diagnosed by increased urinary albumin-to-creatinine ratio (uACR ≥ 30 mg/g) and/or reduced eGFR (eGFR < 60 mL/min/1.73m^2^). However, both measures can vary over time due to intercurrent illness, hydration status and medication changes [2,5], and they have limited accuracy in predicting future decline in renal function [6]. The relationship between uACR and eGFR is also variable, an example being the development of CKD (eGFR < 60 mL/min/1.73 m^2^) without albuminuria [7]. A large number of promising urinary and plasma biomarkers have been assessed in the context of CKD [8] but large-scale longitudinal validation is lacking.

Previous studies using participants from the Fremantle Diabetes Study Phase II (FDS2) identified a novel panel of plasma protein biomarkers associated with CKD in type 2 diabetes that predicted future renal decline independent of known clinical risk factors including uACR and eGFR [9,10]. This led to the development of a predictive diagnostic test for CKD in diabetes known as the PromarkerD test system [11]. The PromarkerD algorithm combines the concentration of three plasma protein biomarkers, apolipoprotein A-IV (apoA4), CD5 antigen-like (CD5L/AIM) and insulin-like growth factor-binding protein 3 (IGFBP3), with the age, serum high-density lipoprotein (HDL)-cholesterol and eGFR of the patient at the time of the PromarkerD test, to provide an estimate of the risk of renal decline within the next four years. The coefficients of the algorithm are adjusted for different definitions of renal decline, and provide prognostic test scores. The PromarkerD test has been validated in two independent cohorts of community-based patients with type 2 diabetes from FDS2 [11], but its potential role in routine clinical practice would be further strengthened by robust external validation.

The aim of the present study was, therefore, to evaluate the PromarkerD test for i) predicting future incident CKD and eGFR decline ≥30% over four years, and ii) diagnosing current CKD at baseline in individuals with type 2 diabetes from the large-scale CANagliflozin CardioVascular Assessment Study (CANVAS), a randomized controlled trial of canagliflozin.

## 2. Experimental Section

### 2.1. Participants

The present study was a post-hoc analysis of data from the CANagliflozin CardioVascular Assessment Study (CANVAS), a randomized controlled trial of canagliflozin versus placebo (ClinicalTrial.gov registration number NCT01032629). Details of the study design and inclusion/exclusion criteria have been previously reported [12]. In brief, the CANVAS program included men and women with type 2 diabetes (glycosylated hemoglobin ≥ 7.0% and ≤10.5%) either ≥30 years of age with a history of cardiovascular disease, or ≥50 years of age with ≥2 risk factors for cardiovascular disease (duration of diabetes mellitus ≥ 10 years, systolic blood pressure > 140 mm Hg while on ≥1 antihypertensive agents, current smoker, documented microalbuminuria or macroalbuminuria, or documented high-density lipoprotein cholesterol <1 mmol/L). All participants were required to have an estimated glomerular filtration rate >30 mL/min/1.73 m^2^ at screening. All patients provided written informed consent, and the trial protocol was approved by the ethics committee at each site [12].

Of the 4330 participants in the modified intention-to-treat (mITT) CANVAS population [12], 3578 (*n* = 1196 placebo arm, *n* = 2382 canagliflozin arm) entered the present PromarkerD sub-study based on informed consent and sample availability. Ten samples failed internal quality checks for data integrity with respect to plasma biomarker concentrations, and were subsequently excluded from further analyses (Figure 1).

### 2.2. Clinical Assessment and Renal Outcomes

All demographic, biochemical and clinical data used in the present study were obtained from the CANVAS trial visit at the time of randomization, denoted as the baseline study visit. The key clinical variables included age, serum HDL-cholesterol, eGFR and uACR. The Chronic Kidney Disease Epidemiology Collaboration (CKD EPI) equation was used for calculating eGFR [13]. The renal outcomes assessed in this study included predicting future decline in renal function during the 4 years from randomization, and confirming CKD at baseline. Future renal decline was defined as (i) incident CKD (eGFR < 60 mL/min/1.73 m^2^ in individuals with an eGFR ≥ 60 mL/min/1.73 m^2^ at randomization), and (ii) eGFR decline ≥ 30% between randomization and follow-up [14]. Diagnosis of CKD at baseline was defined as an eGFR < 60 mL/min/1.73 m^2^ and/or uACR ≥ 30 mg/g [2]. Microalbuminuria and macroalbuminuria were defined as a first-morning urinary ACR ≥ 30 mg/g and >300 mg/g, respectively.

### 2.3. Predicting Renal Status Using PromarkerD Scores

PromarkerD scores were calculated at baseline using a previously defined algorithm to give prognostic (incident CKD and eGFR decline ≥30%, separately) and diagnostic (baseline CKD) test scores [11]. Clinical data (age, serum HDL-cholesterol and eGFR) from the baseline CANVAS trial visit was combined with protein biomarker concentrations (ApoA4, CD5L, and IGFBP3) measured in the present study, to provide PromarkerD prognostic test scores. The diagnostic test score uses the same criteria but does not require baseline eGFR.

Protein biomarkers were measured by immunoaffinity targeted mass spectrometry (MS) using archived baseline plasma samples stored at −80°C. The targeted MS method utilizes bead-based antibody binding for the specific PromarkerD protein biomarkers in a single multiplex capture step. The captured protein biomarkers are reduced, alkylated and digested in situ on the beads with injection onto a microflow liquid chromatography MS system [15]. The PromarkerD scores are predicted probabilities of renal outcomes (incident CKD, eGFR decline ≥30% and baseline CKD, separately) ranging from 0% to 100% and categorized as low-, moderate- or high-risk as determined by pre-specified cut-offs for optimal sensitivity and specificity [11]. For the prognostic scores, the cut-offs were set at 10% and 20%, while the diagnostic score cut-offs were set at 30% and 60%. Participants with prognostic scores <10% were categorized as ‘low’ risk, 10% to <20% as ‘moderate’ risk, and ≥20% as ‘high’ risk. The diagnostic scores were categorized in a similar manner.

### 2.4. Statistical Analyses

Statistical analyses were performed in SPSS Statistics Subscription (build 1.0.0.1327; SPSS Inc., Chicago, IL). A two-tailed level of significance of *p <* 0.05 was used throughout. Data are presented as proportions, mean ± SD, geometric mean (SD range), or, in the case of variables which did not conform to a normal or log_e_-normal distribution (ln), median and inter-quartile range (IQR). All biomarker concentrations were ln-transformed prior to analysis. For independent samples, two-way comparisons for proportions were by Fisher’s exact test, for normally distributed variables by Student’s *t*-test, and for non-normally distributed variables by the Mann–Whitney *U*-test.

Details of the development and validation of the PromarkerD test have been described previously [9,10,11]. Briefly, the linear predictor for each participant was computed using the coefficients from the final logistic regression model and the respective participant data. This was subsequently exponentiated to determine the predicted probability of renal decline. The performance of the PromarkerD scores were assessed against the respective renal outcome in i) participants from the placebo arm only, and ii) participants from the placebo and canagliflozin arms. In order to account for the effect of canagliflozin when all participants were analyzed, a logistic regression model was used to fit each renal outcome against the PromarkerD (prognostic and diagnostic) scores and treatment. The nominal p-value and associated odds ratio (OR) with 95% confidence interval (CI) for each PromarkerD score are reported. This logistic regression modeling was performed using both the continuous score (predicted probability) and risk categories (moderate- or high-risk compared to the low-risk category as reference).

PromarkerD performance was assessed using indices of discrimination and calibration. Model discrimination was assessed by the area under the receiver operating characteristic curve (ROC-AUC) and optimized for differentiating individuals at high risk of renal outcomes from those at low risk, using pre-defined test cut-offs [11]. The sensitivity, specificity, positive predictive value (PPV) and negative predictive value (NPV) were determined at test cut-offs and the maximum Youden Index. Model calibration was determined graphically by plotting observed and predicted numbers of participants who experienced a renal outcome across deciles of risk.

## 3. Results

### 3.1. Baseline Participant Characteristics and Renal Outcomes

Baseline clinical and demographic characteristics of the 3568 participants in the present CANVAS sub-study are summarized in Table 1. The included participants were similar in terms of age, sex, BMI, diabetes duration and renal function to those in the CANVAS mITT cohort. In addition, there were no significant differences in demographic or clinical characteristics between subjects excluded following quality control checks (*n* = 10) and those included in the final analysis (*n* = 3568). The 3568 participants had a mean ± SD age of 62.7 ± 7.9 years, 33% were females and their median (IQR) diabetes duration was 12.4 (8.0–18.0) years. The mean baseline eGFR was 77.0 ± 18.8 mL/min/1.73 m^2^, 16.5% had renal impairment (eGFR <60 mL/min/1.73 m^2^), and 22.1% were microalbuminuric and 5.7% macroalbuminuric. There were 1351 (38.0%) participants with CKD at baseline, defined by a composite of eGFR and uACR (eGFR < 60 mL/min/1.73 m^2^ and/or uACR ≥ 30 mg/g). Excluding those with baseline eGFR < 60 mL/min/1.73 m^2^, during the 4-year follow-up period, 926 (31.1%) individuals developed CKD (*n* = 274 placebo arm, *n* = 652 canagliflozin arm). During follow-up, 564 (16.0%) individuals suffered a decline in eGFR ≥ 30% over four years (*n* = 187 placebo arm, *n* = 377 canagliflozin arm).

### 3.2. Prognostic and Diagnostic PromarkerD test Performance

The descriptive statistics for the PromarkerD scores are provided in Appendix A. There was no significant difference in baseline PromarkerD scores by allocated treatment (*p* = 0.56). The association between PromarkerD scores and the respective renal outcomes are shown in Table 2. After adjusting for the effect of canagliflozin, the prognostic score (modelled as both a continuous and categorical variable) was significantly associated with predicting incident CKD (all *p* ≤ 2.8 × 10^−47^). Moderate-risk and high-risk scores were increasingly prognostic for incident CKD (OR 5.29 (95% CI 4.22–6.64) and OR 13.52 (10.69–17.11) versus low-risk, respectively; both *p* < 2.8 × 10^−47^). Similarly, diagnostic moderate-risk and high-risk scores were increasingly associated with CKD prior to randomization (OR 1.52 (95% CI 1.28–1.80) and OR 2.94 (2.19–3.95) versus low-risk, respectively; both *p* ≤ 1.0 × 10^−6^). The prognostic score was significantly associated with an eGFR decline ≥30% when analyzed as a continuous score (OR 1.13 (95% CI 1.04–1.24), *p* = 5.0 × 10^−3^).

The performance of PromarkerD scores was assessed using ROC-AUC analysis (Table 3). The prognostic score performed the best for predicting incident CKD, providing good discrimination in both placebo-only and all (placebo plus canagliflozin-treated) participants (AUC = 0.79 (95% CI 0.76–0.82) and 0.81 (0.80–0.83), respectively). At the optimal score cut-off, this test score provided acceptable sensitivity (69.3–73.2%), specificity (76.7–76.8%), positive predictive value (53.5–58.8%) and negative predictive value (86.6–86.4%) in predicting four-year risk of developing CKD in both placebo-only and all participants, respectively. At the pre-defined test cut-offs, there were improvements in specificity (93.4–94.0%) for those in the high-risk category, with loss of sensitivity as the cut-off increased (Table 3). The test performed poorly in differentiating participants with rapid eGFR decline (≥30% drop in eGFR) from those with lesser declines in both placebo-only and all participants (AUC = 0.49 (95% CI 0.45–0.54) and 0.54 (0.52–0.57), respectively). The prognostic score underestimated risk for developing CKD and eGFR decline ≥30% across the deciles of risk (Appendix A).

Diagnostic scores were analyzed in participants from both arms of CANVAS, as this was cross-sectional in nature and scores were measured prior to treatment initiation. The diagnostic score discriminated participants with and without CKD (based on a composite of eGFR and uACR), but the performance was low (AUC = 0.58 (95% CI 0.56–0.60)). At the optimal score cut-off, this test score provided 61.2% sensitivity, 50.0% specificity, 42.9% positive predictive value and 67.7% negative predictive value to diagnose baseline CKD. At the pre-defined test cut-offs, there were improvements in sensitivity (81.4%) and specificity (95.2%), to accurately categorize participants as moderate- or high-risk, respectively (Table 3). The diagnostic score underestimated the number of participants with baseline CKD across the lower deciles and overestimated for those in higher risk categories (Appendix A).

## 4. Discussion

The present post-hoc analysis of data from the CANVAS program provides further validation of the PromarkerD test system for predicting future renal decline in type 2 diabetes. Three biomarkers, apoA4, CD5L and IGFBP3, combined with a limited number of routinely available conventional clinical variables, accurately predicted incident CKD during a four-year period. In addition, PromarkerD was able to identify current kidney disease, but had limited ability to predict future eGFR decline ≥30% in CANVAS. This is the first external validation study of PromarkerD which confirms the prognostic utility of the test for incident CKD in a cohort of patients with type 2 diabetes at high-risk of cardiovascular disease.

In the present study, higher PromarkerD scores were significantly predictive of incident CKD during the four-year follow-up period. After adjustment for treatment with canagliflozin, moderate- and high-risk PromarkerD scores were increasingly prognostic for incident CKD (odds ratio 5.29 and 13.52 versus low-risk, respectively). PromarkerD had good utility for predicting incident CKD in terms of discrimination (AUC range 0.79 to 0.81), but calibration was poor reflecting the small numbers of participants in lower deciles. Test performance at the optimal cut-off for predicting incident CKD was lower in CANVAS (AUC 0.81, sensitivity 73%, specificity 77%) compared to FDS2 (the community-based cohort used to develop the test) where the AUC was 0.88, sensitivity 86% and specificity 78% [11]. In CANVAS, PromarkerD had a sensitivity of 61% at the moderate-risk cut-off and specificity of 94% at the high-risk cut-off for predicting incident CKD, which was comparable to that observed in FDS2 (sensitivity 86% and specificity 85%, respectively).

The performance of PromarkerD for predicting an eGFR decline ≥30% during the four-year follow-up period was poor. The AUC for this outcome was nominally significant (AUC 0.54, 95% CI 0.52–0.57), with higher scores predictive of outcome, but only when analyzed as a continuous score (OR 1.13 95% CI 1.04–1.24). Previous studies in FDS2 showed that PromarkerD provided acceptable performance to predict an eGFR decline ≥30% at both development and validation stages (AUC 0.81 and 0.73, respectively) [11]. There are a number of possible reasons for the lower performance observed in the present study. The distribution of PromarkerD scores for the eGFR decline ≥30% outcome was highly skewed, with the majority of CANVAS participants (>98%) having low-risk scores and none having high-risk scores. Given this skewed distribution, there was limited power to detect even small effect sizes for PromarkerD moderate-risk scores versus low-risk scores on outcome.

There are also demographic and clinical differences between CANVAS and FDS2. Subjects in CANVAS had glycated hemoglobin in the range 7.0 to 10.5%, and had either a history of cardiovascular disease or were at high-risk of future cardiovascular events based on risk factors (long diabetes duration, hypertension, current smoking, albuminuria or low serum HDL-cholesterol) [12]. In contrast, FDS2 is a community-based cohort that is representative of patients with type 2 diabetes from an urban Australian population [16]. Compared to FDS2, CANVAS participants at baseline were younger (mean age 62.7 versus 65.5 years), more were male (67.0% versus 54.3%), the median glycated hemoglobin was higher (8.0% versus 6.8%), and the median diabetes duration longer (12.4 versus 7.0 years) [11]. In terms of baseline renal function, CANVAS participants had more renal impairment (16.5% with eGFR < 60 mL/min/1.73 m^2^), but less albuminuria (27.8% with uACR ≥ 30 mg/g) compared to those in FDS2 (11.6% with renal impairment and 34.2% with albuminuria). A detailed assessment of the impact of these demographic and clinical differences on PromarkerD test performance was beyond the scope of the present analyses.

It is not uncommon to observe poor test performance in external validation studies, often due to differences in patient characteristics across different clinical settings, measurement and definition of predictor variables and event rates [17]. Indeed, the definitions of renal decline used in the present study differed to those used in the development of the PromarkerD test, where outcomes were defined at the year 4 study visit only [10,11]. In the present study, there were low event numbers specifically at the year 4 visit, so all events that occurred during the four-year follow-up from randomization were included. Compared to FDS2, event rates were higher in the CANVAS participants over the four-year follow-up period (16–31% versus 7–11%, for incident CKD and eGFR decline ≥30%, respectively), and this may explain the lower discriminative ability of PromarkerD in the present study [11]. Nevertheless, the expansion of the outcome definitions provides additional support for the use of PromarkerD, but further studies to explore this in detail are warranted.

While the strength of PromarkerD lies in its prognostic utility, it also has some confirmatory diagnostic capabilities. In the present study, baseline PromarkerD scores were significantly associated with baseline CKD in CANVAS, defined by a composite of eGFR and uACR (eGFR < 60 mL/min/1.73 m^2^ and/or uACR ≥ 30 mg/g). This definition of CKD diagnosis was chosen based on the current Kidney Disease: Improving Global Outcomes (KDIGO) CKD classification guidelines [2], and allows the detection of albuminuria in the absence of renal impairment (eGFR < 60 mL/min/1.73 m^2^), and renal impairment in the absence of albuminuria, which is highly prevalent in people with diabetes [7]. Moderate-risk and high-risk diagnostic scores were significantly associated with baseline CKD with increasing effect sizes (odds ratio 1.52 and 2.94 versus low-risk, respectively). The discriminative ability of the diagnostic score was low (AUC 0.58 (95% CI 0.56–0.60)), but sensitivity and specificity at the test cut-offs were acceptable (81.4% and 95.2%, respectively).

The present study had a number of strengths including its large sample size and inclusion of people with type 2 diabetes from a multi-center clinical trial independent of the Australian population-based cohort used to develop the test. The participants were well characterized with clinical data available from biannual CANVAS trial visits. The present study also had limitations. The majority of participants were Caucasian (81%), limiting the generalizability of the PromarkerD to other racial and ethnic groups, but a separate analysis in minority groups is underway. Only baseline clinical and biomarker data were used to predict outcomes, and subsequent changes in biomarker concentrations or the effect of canagliflozin on longitudinal PromarkerD scores were not considered.

In conclusion, analysis of the PromarkerD test system in CANVAS shows that the test can predict clinically significant incident CKD in people with type 2 diabetes at high-risk for cardiovascular disease, but was limited in the prediction of rapid decline in eGFR. These data provide external validation of the PromarkerD test for predicting renal decline in type 2 diabetes, but with the caveat that the overall performance observed was less robust than previous studies in community-based people with type 2 diabetes. PromarkerD has the potential to facilitate preventive management strategies which may lead to improved patient care and patient outcomes.

## Figures and Tables

**Figure 1 jcm-09-03212-f001:**
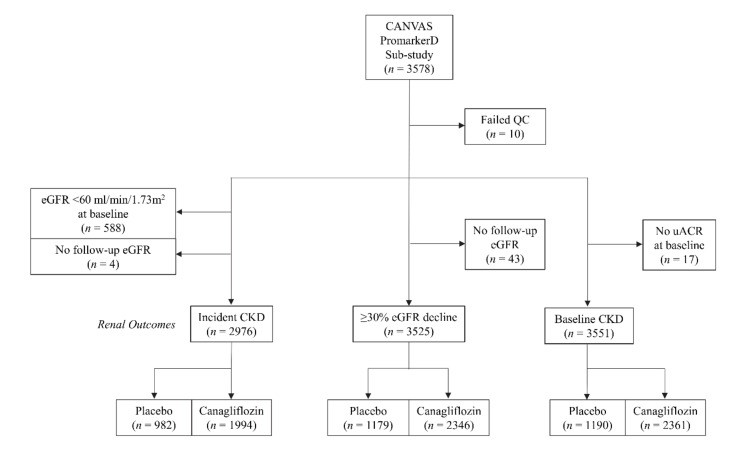
Consort diagram showing the number of CANagliflozin CardioVascular Assessment Study (CANVAS) subjects included in the present PromarkerD sub-study.

**Table 1 jcm-09-03212-t001:** The baseline demographic and clinical characteristics of the 3,568 CANVAS participants used for PromarkerD analysis by treatment arm.

Characteristic	PBO	CANA	Total
Number of samples (%)	1195 (33.5)	2373 (66.5)	3568
Age (years)	62.5 ± 7.8	62.8 ± 7.9	62.7 ± 7.9
Female sex, *n* (%)	394 (33.0)	783 (32.9)	1177 (33.0)
BMI (kg/m^2^)	32.6 ± 6.2	32.7 ± 6.1	32.7 ± 6.1
Diabetes duration (years) *	12.0 (8.0–17.0)	13.0 (8.0–18.0)	12.4 (8.0–18.0)
Fasting plasma glucose (mmol/L) *	9.0 (7.5–10.9)	9.1 (7.6–10.8)	9.0 (7.6–10.9)
HbA_1c_ (%) *	8.0 (7.5–8.7)	8.0 (7.5–8.7)	8.0 (7.5–8.7)
Serum total cholesterol (mmol/L)	4.4 ± 1.2	4.4 ± 1.1	4.4 ± 1.2
Serum HDL-cholesterol (mmol/L)	1.19 ± 0.31	1.20 ± 0.32	1.19 ± 0.32
Serum triglycerides (mmol/L) †	1.7 (1.0–2.9)	1.7 (1.1–2.8)	1.7 (1.0–2.9)
Systolic blood pressure (mmHg)	137 ± 16	136 ± 16	137 ± 16
Diastolic blood pressure (mmHg)	78 ± 10	77 ± 10	78 ± 10
Diuretic use, *n* (%)	555 (46.4)	1101 (46.4)	1656 (46.4)
History of heart failure, *n* (%)	173 (14.5)	298 (12.6)	471 (13.2)
Urine albumin to creatinine ratio (mg/g) *	11.6 (6.2–37.1)	11.7 (6.5–34.6)	11.7 (6.4–35.6)
eGFR (mL/min/1.73m^2^)	76.8 ± 18.9	77.1 ± 18.7	77.0 ± 18.8
eGFR < 60 mL/min/1.73m^2^, *n* (%)	212 (17.7)	376 (15.8)	588 (16.5)

All values are mean ± SD (standard deviation) unless labeled otherwise; * Median (IQR—interquartile range); † Geometric Mean (SD range). PBO, placebo; CANA, canagliflozin; BMI, body mass index; eGFR, estimated glomerular filtration rate by chronic kidney disease (CKD) Epidemiology Collaboration equation; HbA_1c_, glycated hemoglobin; HDL, high-density lipoprotein.

**Table 2 jcm-09-03212-t002:** Association of PromarkerD Scores (continuous and risk categories) with renal outcomes in CANVAS participants.

PromarkerD Score (Outcome)	Treatment Arm	Number with Outcome/Total (%)	Odds Ratios (95% CI), *p*-value		
			Continuous	Risk Category (Mod vs. Low)	Risk Category (High vs. Low)
Prognostic * (Incident CKD)	PBO	274/982 (27.9%)	1.85 (1.67–2.06) 2.0 × 10^−31^	4.04 (2.75–5.93) 1.1 × 10^−12^	10.78 (7.21–16.12) 5.0 × 10^−31^
	PBO + CANA	926/2976 (31.1%)	2.02 (1.90–2.15) 2.3 × 10^−109^	5.29 (4.22–6.64) 2.8 × 10^−47^	13.52 (10.69–17.11) 1.3 × 10^−104^
Prognostic * (Decline ≥30%)	PBO	187/1179 (15.9%)	0.98 (0.85–1.13) 0.76	0.64 (0.23–1.74) 0.38	N/A ^†^
	PBO + CANA	564/3525 (16.0%)	1.13 (1.04–1.24) 5.0 × 10^−3^	1.73 (0.96–3.12) 0.068	N/A ^†^
Diagnostic (Baseline CKD)	PBO + CANA	1351/3551 (38.0%)	1.02 (1.01–1.02) 4.3 × 10^−14^	1.52 (1.28–1.80) 1.0 × 10^−6^	2.94 (2.19–3.95) 9.9 × 10^−13^

* The two prognostic scores were ln-transformed due to a non-normal distribution. A 2.72-fold increase in prognostic score corresponds to an increase of 1 in ln (prognostic score). Only participants with PromarkerD scores and outcome data are shown. Odds ratios were adjusted for canagliflozin treatment when subjects from both placebo (PBO) and canagliflozin (CANA) arms were included in the analysis. No adjustment for treatment was made for the diagnostic scores. ^†^ No participants were observed in the prognostic (decline ≥30%) high-risk category.

**Table 3 jcm-09-03212-t003:** Test performance of the PromarkerD scores in CANVAS participants.

Participants	PromarkerD Scores				
	Prognostic (Incident CKD)		Prognostic (Decline ≥30%)		Diagnostic (Baseline CKD)
	PBO	PBO + CANA	PBO	PBO + CANA	PBO + CANA
Number of Subjects	982	2976	1179	3525	3551
Observed outcomes (%)	274 (27.9%)	926 (31.1%)	187 (15.9%)	564 (16.0%)	1351 (38.0%)
AUC (95%CI)	0.79 (0.76–0.82)	0.81 (0.80–0.83)	0.49 (0.45–0.54)	0.54 (0.52–0.57)	0.58 (0.56–0.60)
At max YI cut-off:	(7.1%)	(5.9%)	(5.6%)	(2.8%)	(38.9%)
Sensitivity (%)	69.3	73.2	16.6	45.9	61.2
Specificity (%)	76.7	76.8	90.0	62.4	50.0
PPV (%)	53.5	58.8	23.8	18.9	42.9
NPV (%)	86.6	86.4	85.1	85.8	67.7
At moderate-risk cut-off *:					
Sensitivity (%)	59.1	60.6	2.7	2.7	81.4
Specificity (%)	81.8	84.6	98.3	98.4	26.9
PPV (%)	55.7	64.0	22.7	24.6	40.6
NPV (%)	83.8	82.6	84.3	84.2	70.2
At high-risk cut-off *:					
Sensitivity (%)	35.8	37.7	N/A^†^	N/A ^†^	9.7
Specificity (%)	93.4	94.0			95.2
PPV (%)	67.6	73.9			55.5
NPV (%)	79.0	77.0			63.2

Only participants with complete data were included in each model. Discrimination performance measures are given for moderate- (10%), and high- (20%) risk cut-offs for 4-year renal decline, as well as for the optimal cut-off (shown in parentheses) defined by maximum Youden Index (YI). * For the Diagnostic Score the cut-offs are 30% (moderate) and 60% (high), and all participants from PBO and CANA arms were included. † No participants were observed in the prognostic (decline ≥30%) high-risk category. PBO, placebo; CANA, canagliflozin; AUC = area under the curve; PPV = positive predictive value; NPV = negative predictive value.

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
