# Peer review of "PromarkerD Predicts Renal Function Decline in Type 2 Diabetes in the Canagliflozin Cardiovascular Assessment Study (CANVAS)"

_jcm, 2020, doi:10.3390/jcm9103212_

Round 1

Reviewer 1 Report

Dear editor,
I have read with interest the paper entitled “”.
The paper is nicely crafted and reads well. However, a few aspects deserve clarification in the reviewer’s opinion:
- Was there any major difference in demographic or clinical characteristics between patients included vs excluded form this post hoc analysis?
- The definition of “incident CKD” is misleading since a substantial portion of patients had CKD (either reduced GFR or proteinuria) at baseline. Authors may want to consider rephrasing it as “future renal function decline” as used to define the renal outcomes on page 3
- Table 1 reports on baseline characteristics of the study cohort according to mITT. However, 10 patients failed urinary QC and should be excluded.
- How did canagligflozin interact with PromarkerD score risk prediction? it is unclear the reasons for reporting in table 2 the association of PromarkerD scores and renal outcome in placebo and not canagliflozin treated patients. Can authors also provide the readers with the association of PromarkerD scores and renal outcomes?
- similarly, it would be of interest to know the performance of PrimorkerD score in canagliflozin treated patients (table 3)

Author Response

Point 1: Was there any major difference in demographic or clinical characteristics between patients included vs excluded from this post hoc analysis?

Response: There was no major difference in any of the demographic or clinical characteristics between patients included versus excluded in the present study. As stated in the second sentence on lines 147-149 of the results, “The included participants were similar in terms of age, sex, BMI, diabetes duration and renal function to those in the CANVAS mITT cohort (data not shown).” However, we have now added a sentence addressing Point 3 of the Reviewer’s comments comparing subjects excluded due to failed quality control checks with those included in the final analysis. Please see tracked changes in the revised manuscript.

Point 2: The definition of “incident CKD” is misleading since a substantial portion of patients had CKD (either reduced GFR or proteinuria) at baseline. Authors may want to consider rephrasing it as “future renal function decline” as used to define the renal outcomes on page 3.

Response: We are unclear what is misleading regarding our current definition of incident CKD. It is clearly defined from the outset in the Abstract on line 21 “incident CKD (defined as estimated glomerular filtration rate (eGFR) <60 mL/min/1.73m2) during the 4 years from randomization, in those with baseline eGFR ≥60 mL/min/1.73m2)”, and again in the Methods on lines 94-98 “The renal outcomes assessed in this study included predicting future decline in renal function during the 4 years from randomization, and confirming CKD at baseline. Future renal decline was defined as i) incident CKD (eGFR <60 mL/min/1.73m2 in individuals with an eGFR ≥60 mL/min/1.73m2 at randomization), and ii) eGFR decline ≥30% between randomization and follow-up [14].” However, in order to improve the clarity of the Abstract, we have changed the second sentence on lines 17-18 to “This study investigated the prognostic utility of a novel blood test, PromarkerD, for predicting future renal function decline in individuals with type 2 diabetes from the CANagliflozin CardioVascular Assessment Study (CANVAS)”. We have also changed the sentence on lines 21-24 of the Abstract to include the other definition of renal function decline assessed in the present study ‘…used to predict incident CKD (estimated glomerular filtration rate (eGFR) <60 mL/min/1.73m2 during follow-up in those above this threshold at baseline) and eGFR decline ≥30% during the 4 years from randomization.” Please see tracked changes in the revised manuscript.

Point 3: Table 1 reports on baseline characteristics of the study cohort according to mITT. However, 10 patients failed urinary QC and should be excluded.

Response: The Reviewer is correct that plasma biomarker concentrations for 10 subjects failed quality control checks. We have removed these 10 subjects from Table 1 and updated the entire table accordingly (line 160). As a result, we have updated some numbers in the Abstract (line 20 – “…at baseline in 3,568 CANVAS participants (n=1,195 placebo arm, n=2,373 canagliflozin arm”) as well as the first paragraph of the Results section (lines 146 and 151-152) and line 255 in the Discussion (“…more were male (67.0% versus 54.3%”). We have also tested if there were any significant differences in demographic or clinical characteristics between subjects excluded following quality control checks and those included in the final analysis, and have added a sentence describing this on line 149. “In addition, there were no significant differences in demographic or clinical characteristics between subjects excluded following quality control checks (n=10) and those included in the final analysis (n=3,568) (data not shown).” Please see tracked changes in the revised manuscript.

Point 4: How did canagliflozin interact with PromarkerD score risk prediction? it is unclear the reasons for reporting in table 2 the association of PromarkerD scores and renal outcome in placebo and not canagliflozin treated patients. Can authors also provide the readers with the association of PromarkerD scores and renal outcomes?

Response: Table 2 shows the association of PromarkerD with outcomes in participants from the placebo arm of the CANVAS trial and in the whole cohort including participants from both the placebo and canagliflozin arms. The difference between the two results is that in the latter, the known effect of canagliflozin on renal outcomes was adjusted for in the models, with the odds ratio (95% confidence intervals) presented adjusted for treatment. This is stated in the footnote to Table 2 (line 183) and on line 130 in the Methods section “In order to account for the effect of canagliflozin when all participants were analyzed, a logistic regression model was used to fit each renal outcome against the PromarkerD (prognostic and diagnostic) scores and treatment.” We have not presented data on the interaction between baseline PromarkerD scores and treatment in the present study, but additional analyses including a PromarkerD*treatment interaction term in models was not found to be significant. It would not be appropriate to show the association of PromarkerD scores with outcomes in participants from the canagliflozin-only arm without adjustment for treatment, and as such, we believe adding these data to the present study in Table 2 or Table 3 would be beyond the scope of the present analysis.

Point 5: similarly, it would be of interest to know the performance of PromarkerD score in canagliflozin treated patients (table 3).

Response: Similar to our response to Point 4 above, we do not believe that providing this data in the present study adds significant value to the manuscript.

Reviewer 2 Report

Present study attempted to validate a novel PromarkerD score in an external cohort of CANVAS study trial. The study shows the association between diagnostic and prognostic scores and prevalent and incident CKD, respectively. The discrimination tests show excellent specificity and acceptable sensitivity for CKD, however failed to predict rapid decline in eGFR. Overall, the results showed poorer performance in the CANVAS cohort compared to previous studies in the community-based cohort. The later is not fully disclosed in the conclusion.   

The additional value of PromarkerD as diagnostic test of CKD defined by routine clinical test is a moot point. More precise evaluation of kidney function and histological damage would be required to prove the diagnostic utility of PromarkerD.  

PromarkerD prognostic score includes baseline eGFR, a principal diagnostic criterion for CKD. Authors may want to provide the prediction utility of Promarker D versus eGFR alone and combination of routine clinical data/tests – age, eGFR and HDL.

PromarkerD score was developed in relatively healthier FDS2 population. The incidence of both CKD and eGFR decline was considerably lower (5 to 10% in about 4 years) compared to 15-30% in the CANVAS. The higher rate of outcomes may explain lower discrimination sensitivity and PPR in the present study. Perhaps using more stringent outcome criteria would improve the score performance. Form clinical point of view, eGFR < 30 or KDIGO stage 4 may be more relevant in patients with more advanced diabetes.

Author Response

Point 1: Present study attempted to validate a novel PromarkerD score in an external cohort of CANVAS study trial. The study shows the association between diagnostic and prognostic scores and prevalent and incident CKD, respectively. The discrimination tests show excellent specificity and acceptable sensitivity for CKD, however failed to predict rapid decline in eGFR. Overall, the results showed poorer performance in the CANVAS cohort compared to previous studies in the community-based cohort. The latter is not fully disclosed in the conclusion.

Response: We agree with the Reviewer’s comments and have rewritten the conclusion accordingly (lines 294-300). “In conclusion, analysis of the PromarkerD test system in CANVAS shows that the test can predict clinically significant incident CKD in people with type 2 diabetes at high-risk for cardiovascular disease, but was limited in the prediction of rapid decline in eGFR. These data provide external validation of the PromarkerD test for predicting renal decline in type 2 diabetes, but with the caveat that the overall performance observed was less robust than previous studies in community-based people with type 2 diabetes. PromarkerD has potential to facilitate preventive management strategies which may lead to improved patient care and patient outcomes.” We have also amended the last sentence of the Abstract “Analysis of the PromarkerD test system in CANVAS show the test can predict clinically significant incident CKD in this multi-center clinical study, but had limited utility for predicting eGFR decline ≥30%.” Please see tracked changes in the revised manuscript.

Point 2: The additional value of PromarkerD as diagnostic test of CKD defined by routine clinical test is a moot point. More precise evaluation of kidney function and histological damage would be required to prove the diagnostic utility of PromarkerD.

Response: While this statement is true, renal biopsies are not usually performed in a usual care setting and would be ethically challenging to justify in the context of a validation study. We acknowledge that the strength of PromarkerD lies in its prognostic utility (as explicitly stated in the Discussion, line 274), but feel that there is some confirmatory diagnostic capability worth including in the present study, namely the fact that using a combination of both eGFR and uACR allows the detection of CKD in people who may have normal kidney function by one measure but not the other, and may therefore be missed if only one test were used.

Point 3: PromarkerD prognostic score includes baseline eGFR, a principal diagnostic criterion for CKD. Authors may want to provide the prediction utility of PromarkerD versus eGFR alone and combination of routine clinical data/tests – age, eGFR and HDL.

Response: The Reviewer has a valid point, but this was not the intention of the present study. The objective of the present study was to assess the prognostic utility of PromarkerD in an external cohort of people with type 2 diabetes that was independent to the original development and validation studies performed using the Fremantle Diabetes Study resources. We have previously shown that the PromarkerD biomarkers add significant independent and incremental prognostic value to a full range of clinical predictors, including both eGFR and urinary albumin-to-creatinine ratio, to predict renal outcomes (Peters et al., 2017: Identification of Novel Circulating Biomarkers Predicting Rapid Decline in Renal Function in Type 2 Diabetes: The Fremantle Diabetes Study Phase II. Diabetes Care 2017, 40, 1548-1555).

Point 4: PromarkerD score was developed in relatively healthier FDS2 population. The incidence of both CKD and eGFR decline was considerably lower (5 to 10% in about 4 years) compared to 15-30% in the CANVAS. The higher rate of outcomes may explain lower discrimination sensitivity and PPR in the present study. Perhaps using more stringent outcome criteria would improve the score performance. Form clinical point of view, eGFR < 30 or KDIGO stage 4 may be more relevant in patients with more advanced diabetes.

Response: The Reviewer is correct that PromarkerD was developed in a relatively healthy population of people with well-controlled type 2 diabetes. We have acknowledged and discussed the key demographic and clinical differences between the FDS2 and CANVAS cohorts in the fourth paragraph of the Discussion section (starting line 249). The differences in event rates may explain the lower sensitivity and specificity of PromarkerD in the CANVAS cohort. We have added a sentence with a comparison of the number of events between the two cohorts to the fifth paragraph of the Discussion at lines 268-271, and adjusted the start of the subsequent sentence accordingly. “Compared to FDS2, event rates were higher in the CANVAS participants over the four-year follow-up period (16-31% versus 7-11%, for incident CKD and eGFR decline ≥30%, respectively), and this may explain the lower discriminative ability of PromarkerD in the present study [11].” Please see tracked changes in the revised manuscript.

The objective of the present study was to assess the prognostic performance of PromarkerD for the intended outcomes, i.e. incident CKD and an eGFR decline of ≥30%. These definitions are clinically useful and allow the target population that may benefit from PromarkerD testing to include the majority of people with type 2 diabetes. We acknowledge that there are a range of other clinically useful definitions of renal decline which may be more relevant to people with more advanced diabetes. We are currently investigating the utility of PromarkerD for predicting a range of additional outcomes.

Round 2

Reviewer 1 Report

Authors have addressed all comments appropriately. The reviewer suggests adding to the methods or result session that "PromarkerD*treatment interaction term in models was not found to be significant" as stated in their rebuttal.

Author Response

Point 1: The reviewer has suggested adding to the methods or result section that "PromarkerD*treatment interaction term in models was not found to be significant" as stated in their rebuttal.

Response: We have added the following sentence to the Results section (lines 178-180) "A PromarkerD score-by-treatment interaction term was included in each model, with no significant effect found (data not shown)."

Reviewer 2 Report

The revision answered all my concerns.

Author Response

There were no revisions requested by this reviewer.